



# Influences of downward transport and photochemistry on surface ozone over East Antarctica during austral summer: in situ observations and model simulations

Imran A. Girach[1,2*], Narendra Ojha[3], Prabha R. Nair[4,2], Kandula V. Subrahmanyam[5,2],
Neelakantan Koushik[6], Mohammed M. Nazeer[6], Nadimpally Kiran Kumar[6], Surendran Nair
Suresh Babu[6], Jos Lelieveld[7], and Andrea Pozzer[7,8*]

[1]Space Applications Centre, Indian Space Research Organisation, Ahmedabad 380015, India
[2]*Formerly at* Space Physics Laboratory, Vikram Sarabhai Space Centre, Thiruvananthapuram 695022, India
[3]Space and Atmospheric Sciences Division, Physical Research Laboratory, Ahmedabad 380009, India
[4]TC 95/1185, Aiswarya Gardens, Kumarapuram, Thiruvananthapuram - 695011, India
[5]National Remote Sensing Centre, Indian Space Research Organisation, Hyderabad 500015, India
[6]Space Physics Laboratory, Vikram Sarabhai Space Centre, Thiruvananthapuram 695022, India
[7]Department of Atmospheric Chemistry, Max Planck Institute for Chemistry, Mainz 55128, Germany
[8]Climate and Atmosphere Research Center, The Cyprus Institute, Nicosia, Cyprus

*Correspondence to*: Imran A. Girach (imran.girach@gmail.com) and Andrea Pozzer
(andrea.pozzer@mpic.de)

**Abstract.** Studies of atmospheric trace gases in remote, pristine environments are critical for assessing the accuracy of climate models and advancing our understanding of natural processes and global changes. We investigated the surface ozone ($O_3$) variability over East Antarctica during the austral summer of 2015–2017 by combining surface and balloon-borne measurements at the Indian station Bharati (69.4° S, 76.2° E, ~35 m above mean sea level) with EMAC atmospheric chemistry-climate model simulations. The model reproduced the observed surface $O_3$ level (18.8 ± 2.3 nmol mol$^{-1}$) with negligible bias and captured much of the variability (R=0.5). Model simulated tropospheric $O_3$ profiles were in reasonable agreement with balloon-borne measurements (mean bias: 3–11 nmol mol$^{-1}$). Our analysis of a stratospheric tracer in the model showed that about 40–50% of surface $O_3$ over the entire Antarctic region was of stratospheric origin. Events of enhanced $O_3$ (~4–10 nmol mol$^{-1}$) were investigated by combining $O_3$ vertical profiles and air mass back trajectories, which revealed the rapid descent of $O_3$-rich air towards the surface. The photochemical loss of $O_3$ through its photolysis (followed by $H_2O+O(^1D)$) and reaction with hydroperoxyl radicals ($O_3+HO_2$) dominated over production from precursor gases ($NO+HO_2$ and $NO+CH_3O_2$) resulting in overall net $O_3$ loss during the austral summer. Interestingly, the east coastal region, including the Bharati station, tends to act as a stronger chemical sink of $O_3$ (~190 pmol mol$^{-1}$ d$^{-1}$) than adjacent land and ocean regions (by ~100 pmol




mol$^{-1}$ d$^{-1}$). This is attributed to reverse latitudinal gradients between $H_2O$ and $O(^1D)$, whereby $O_3$ loss through photolysis ($H_2O+O(^1D)$) reaches a maximum over the east coast. Further, the net photochemical loss at the surface is counterbalanced by downward $O_3$ fluxes, maintaining the observed $O_3$ levels. The $O_3$ diurnal variability of ~1.5 nmol mol$^{-1}$ was a manifestation of combined effects of mesoscale wind changes and up- and downdrafts, in addition to the net photochemical loss. The study provides valuable insights into the intertwined dynamical and chemical processes governing the $O_3$ levels and variability over East Antarctica.

## 1    Introduction

Tropospheric ozone ($O_3$) plays a pivotal role in governing the atmospheric oxidation capacity, and influences air quality and climate warming (Seinfeld and Pandis, 1998). The major source of $O_3$ in the troposphere is its photochemical formation involving precursors such as nitrogen oxides ($NO_x$), carbon monoxide (CO) and non-methane hydrocarbons (NMHCs; Lelieveld and Dentener, 2000). The contribution of downward transport from the stratosphere is generally minor near the surface, although it can be significant at middle to high latitudes (Stohl et al., 2003). Numerous studies have investigated the chemistry and dynamics of tropospheric $O_3$ and the roles of local to synoptic-scale processes (e.g., boundary layer height variation, horizontal and vertical transport, etc.; Nguyen et al., 2022; Young et al., 2018). Investigations of $O_3$ variations in remote pristine environments, isolated from major anthropogenic influences, are essential to understand the global changes in atmospheric composition, the role of natural processes including downward transport from the stratosphere, and photo-denitrification of the snowpack (Jones et al., 2001). In this regard, the observations over environments such as Antarctica are extremely valuable and can provide insights into the global background atmosphere, besides providing data to test the results of chemistry-climate models. The mean surface $O_3$ over the Antarctic region was observed to be lower by nearly 5 nmol mol$^{-1}$ than that over the Arctic polar region (Helmig et al., 2007). Surface $O_3$ shows a pronounced seasonality (~15–20 nmol mol$^{-1}$ amplitude) with a summer minimum and a winter maximum over Antarctica, accompanied by periodic fluctuations associated with long-range transport (Kumar et al., 2021; Legrand et al., 2016; Oltmans and Komhyr, 1976; Winkler et al., 1992). In line with global increases in tropospheric $O_3$ due to the enhanced anthropogenic emissions since the pre-industrial era and impacts of climate warming





(Wang et al., 2022; Murazaki and Hess, 2006; Lelieveld et al., 2004), an increasing trend (<0.2 nmol mol$^{-1}$ y$^{-1}$) in surface $O_3$ has also been reported from Antarctica (Kumar et al., 2021).

Previous studies have investigated the long-term, inter-annual, seasonal and diurnal variations in surface $O_3$ over Antarctica (Legrand et al., 2009, 2016), as well as the role of horizontal transport (Tian et al., 2022) and chemistry including that of radicals (Preunkert et al.,
2012), halogen driven $O_3$ depletion (Tarasick and Bottenheim, 2002; Jones et al., 2013), and stratospheric intrusions (Das et al., 2020). Antarctic observations have provided evidence of widespread $O_3$ production during austral spring and summertime, affecting all stations through horizontal mixing. This $O_3$ production contributes to a significant enhancement in annual mean $O_3$ over the Antarctic plateau (Helmig et al., 2007). While a weak coupling between stratospheric and
tropospheric $O_3$ was inferred earlier (Oltmans and Komhyr, 1976), frequent stratospheric intrusions in this region were also reported (Cristofanelli et al., 2018; Das et al., 2020; Greenslade et al., 2017). There have been extensive studies on a range of species utilizing datasets from dedicated campaigns and projects over West Antarctica and South Pole (CHABLIS—Chemistry of the Antarctic Boundary Layer and the Interface with Snow, Jones et al., 2008;
ISCAT—Investigation of Sulfur Chemistry in Antarctica, Davis et al., 2004; ANTCI—Antarctic Tropospheric Chemistry Investigation (Eisele et al., 2008); WAIS—West Antarctic Ice Sheet, Frey et al., 2005; Masclin et al., 2013). The variability of volatile organic compounds (VOCs), radicals, $O_3$ and its precursors has been investigated over the East Antarctic plateau and eastern coastal Antarctica (OPALE—Oxidant Production over Antarctic Land and its Export; Preunkert et al.,
2012 and references therein). But the east coast of Antarctica remains a relatively less explored region as compared to west Antarctica and the South Pole.

The east coast of Antarctica is distinct from the west coast as well as the inland region of Antarctica. Relatively high levels of hydroxyl and peroxy radicals over eastern Antarctica (Dumont d'Urville; 66.67° S, 140.02° E, 40 m above mean sea level—amsl) during austral summer
(Kukui et al., 2012) indicate chemical differences from west Antarctica (Palmer; 64.77° S; 64.05° W) where radical concentrations are lower. Short-term events of $O_3$ enhancements are observed over the coastal as well as inland regions with higher frequency during the summer season and they are associated with ultraviolet radiation reaching the surface, photochemical production and transport (Crawford et al., 2001; Frey et al., 2015; Cristofanelli et al., 2018; Legrand et al., 2016).



Net summertime $O_3$ production (4–5 nmol mol$^{-1}$ d$^{-1}$) has been observed at the eastern coastal Antarctica through $NO_x$ emission from snow (Legrand et al., 2009, 2016). In contrast, surface or boundary layer $O_3$ depletion are also observed mainly due to halogen chemistry involving iodine, bromine and chlorine oxides, being more frequent in west Antarctica (Saiz-Lopez et al., 2007; Simpson et al., 2007). Weaker or less frequent surface $O_3$ depletion is observed over the east coast compared to the west coast of Antarctica (Jones et al., 2013; Legrand et al., 2016).

Most studies of East Antarctica have been based on in situ measurements of various trace gases including radical species ($O_3$, NO, HONO, OH, DMS, BrO, etc.) at Dumont d'Urville, Syowa (69.00° S; 39.58° E, ~29 m amsl) and Zhongshan (69.37° S, 76.36° E, 18.5 m amsl; Kukui et al., 2012; Legrand et al., 2016, 2009; Murayama et al., 1992; Tian et al., 2022) stations. These studies have shown the surface $O_3$ variability on different scales (i.e., diurnal: ~2 nmol mol$^{-1}$, seasonal: ~18 nmol mol$^{-1}$, and long-term trend: 0.07±0.07 nmol mol$^{-1}$ y$^{-1}$). Only few studies have analyzed the relevant larger-scale trace gas distributions and discussed the model performance of seasonal changes in surface $O_3$ or tropospheric $O_3$ (Wang et al., 2022; Griffiths et al., 2021), including halogen chemistry (Yang et al., 2005; Fernandez et al., 2019). Studies investigating the chemistry and dynamics of surface $O_3$ are scarce for Antarctica (Morgenstern et al., 2013). To the best of our knowledge, there are no comprehensive studies discussing the surface $O_3$ variability and associated processes based on the synergy of in situ measurements and chemistry-climate modeling over East Antarctica. It is timely to investigate the underlying processes since an increasing $O_3$ trend has been reported over this part of the world recently (Kumar et al., 2021).

Our study aims to contribute to the understanding of chemical and dynamical processes governing the surface $O_3$ variability over the east coast of Antarctica. We have conducted in-situ measurements during three different years and performed simulations using a global chemistry-climate model to unravel the atmospheric processes that control the summertime $O_3$ levels and variability. Details of the measurements and model simulations are given in the next section. Results of the $O_3$ variability and a comparison of model results with measurements, and an analysis of photochemical and dynamical contributions are presented in section 3. A summary, the main conclusions, and a future outlook are presented in section 4.



## 2 Measurements and model simulations

### 2.1 In situ measurements


Surface $O_3$ was measured at the Indian station Bharati (69.4° S, 76.2° E, ~35 m above mean sea level) at the Larsemann Hills in the east coast of Antarctica during the summer seasons of three years 2015–2017: 29 January–13 February 2015, 17 January–24 February 2016, and 11 December 2016–16 February 2017. The Bharati site experiences a surface pressure of ~980±10 hPa, cold

temperatures (-0.1±3 °C; -11–8 °C), moderate humidity (60±13.5%; 34–98%) and mainly easterly winds with a number of blizzards during the summer season. A detailed overview of the meteorological conditions at Bharati station can be found in Soni et al., 2017.

Surface $O_3$ mixing ratios were measured using an online ultraviolet photometric ozone analyzer manufactured by the Environnement S.A, France (model O342). The instrument derives

$O_3$ mixing ratios using the Beer–Lambert law considering the absorption of ultraviolet radiation around 253.7 nm by $O_3$ molecules. The measurement noise, lower detection limits, linearity and minimum response time are 0.5 nmol mol$^{-1}$, 1 nmol mol$^{-1}$, ±1 % and 10 s, respectively. The instrument was operated on the auto-response mode (response time of 10–90 s) under permissible range of temperature. $O_3$ mixing ratios were recorded continuously at 5 min averaging intervals.

Air samples were drawn from a height of approximately 2 m above the ground level through a Teflon tube and filtered through 5 µm non-reactive polytetrafluoroethylene dust filter prior to injection into the analyzer. Prior to each expedition, the analyzer was calibrated for mixing ratios of 20 and 30 nmol mol$^{-1}$ using a multichannel calibrator. The measurement uncertainty is estimated to be ~5 % (Tanimoto et al., 2007). In addition to measurements at Bharati, surface $O_3$ at Syowa

and Arrival Heights (77.80° S; 166.67° E) available from https://ebas-data.nilu.no/Default.aspx for the study period are also used for the comparison of model results.

The vertical profiles of $O_3$ partial pressure were measured using balloon-borne electrochemical ozonesondes manufactured by the En-Sci Corporation, USA (Model: 2Z-V7). A total of 12 profiles were measured during the study period. The $O_3$ partial pressure was converted

to $O_3$ mixing ratios using the simultaneously measured atmospheric pressure by radiosonde (model: iMet-1-RSB). Air is passed through an electrochemical concentration cell (ECC) using a built-in non-reactive pump, and the current generated by the electrochemical reaction of $O_3$ (with



potassium iodide) is measured by an electronic interface board and converted into an $O_3$ partial pressure. The detailed operation principle and performance evaluation of ozonesonde instrument

are described in Komhyr et al., 1995 and references therein. The accuracy of $O_3$ measurements is reported to be 5–10% up to an altitude of 30 km (Smit et al., 2007). Additional details of the $O_3$ measurements and meteorological parameters using this technique can be found elsewhere (Ajayakumar et al., 2019; Ojha et al., 2014). Besides our measurements at Bharati, we utilized available $O_3$ vertical profiles measured using ECC ozonesondes at Davis station (68.58° S 77.97°

E; https://data.aad.gov.au/metadata/records/AAS_4293_Ozonesonde) in this study.

The surface level wind speed and direction were measured using an automatic weather station, which meets the standards of the World Meteorological Organization and was operated by the India Meteorological Department. Wind direction measurements are used here to analyze the changes in surface $O_3$ on a diurnal time scale. To understand the impacts of updraft and downdrafts,

the vertical wind at the surface was measured using a fast response ultrasonic anemometer (make: METEK, GmBH, Germany; model: USA-1 Scientific). The factory calibrated sensor was mounted at 3 m level above the ground and was operated at 25 Hz during January 2016. The measuring resolution and accuracy of the vertical velocity are $\pm\,0.01$ m s$^{-1}$ and 0.2 m s$^{-1}$, respectively. Further details on the instrument can be found in (Reddy et al., 2021)

**2.2    Model simulations**

In this work the EMAC (ECHAM5/MESSy Atmospheric Chemistry) model (Jöckel et al., 2010, 2006) has been used. This model is a numerical chemistry and climate simulation system that includes sub-models describing tropospheric and middle atmospheric processes and their interaction with oceans, land and human influences. It uses the second version of the Modular

Earth Submodel System (MESSy2) to link multi-institutional computer codes. The core atmospheric model is the 5th generation European Centre Hamburg general circulation model (ECHAM5, Roeckner et al., 2006). The physics subroutines of the original ECHAM code have been modularized and reimplemented as MESSy submodels and have continuously been further developed. Only the spectral transform core, the flux-form semi-Lagrangian large scale advection

scheme, and the nudging routines for Newtonian relaxation are remaining from ECHAM5. For the present study we applied EMAC (MESSy version 2.55.0) in the T106L47MA-resolution, i.e. with





a spherical truncation of T106 (corresponding to a quadratic Gaussian grid of approximately 1.1° × 1.1° in latitude and longitude) with 47 vertical hybrid pressure levels up to 0.01 hPa. In this work we used the same set up as in Reifenberg et al. (2022), and the model results encompass the

years 2014-2018 with a 3-hours output frequency. Global atmospheric chemistry models are known to overestimate tropospheric ozone (Young et al., 2018), and EMAC is no exception to this. Nevertheless, extensive ozone evaluation (Jöckel et al., 2016) shows that the EMAC model has a very low (less than 10%) or no bias in the troposphere against observations for latitudes below 60° S. Furthermore, the EMAC model has been extensively evaluated in the last years both

for the gas phase (Jöckel et al., 2016; Taraborrelli et al., 2021) and for the aerosol phase (Pozzer et al., 2012; Brühl et al., 2018; Pozzer et al., 2022).

To investigate the effects of transport, airmass back trajectories have been computed using the HYSPLIT (HYbrid Single Particle Lagrangian Integrated Trajectory) model version-4 (Rolph et al., 2017; Stein et al., 2015) with the input of 1°×1° gridded GDAS (Global Data Assimilation

System) meteorological data.

## 3    Results and Discussions

### 3.1    O₃ variability: comparison of observations with model simulations

Figure 1a shows the elevation map of Antarctica marked with the location of the Indian station Bharati (69.4° S, 76.2° E, ~35 m amsl), where surface-based and balloon-borne

measurements of $O_3$ have been conducted during this study. The surface elevation is higher (up to 4 km) over the eastern part of Antarctica. Figure 1b shows the spatial distribution of surface $O_3$ during summer of 2015–2017 (29 January–13 February 2015, 17 January–24 February 2016, and 11 December 2016–16 February 2017) as simulated by the EMAC model, along with the mean observed value at Bharati station ($18.8 \pm 2.3$ nmol mol$^{-1}$). The mean $O_3$ distribution shows increase

from the oceanic region (10–16 nmol mol$^{-1}$) to the landmass (15–23 nmol mol$^{-1}$), nearly following the topographical features of Antarctica. Overall, the model simulated spatial distribution of $O_3$ (Fig. 1b) is seen to be in agreement with the distribution based on measurements from different stations (Fig. S1). This is further consistent with previous studies showing higher $O_3$ mixing ratios over elevated sites (Summit; 3212 m amsl and South Pole; 2830 m amsl) as compared to the





coastal/oceanic region (Helmig et al., 2007). The balloon-borne observations (Fig. 3) also show
       increase in mean $O_3$ mixing ratios with altitude.

       Figure 1c shows the stratospheric contribution (in percent) to the surface $O_3$ based on the
       stratospheric $O_3$ tracer in the model ($O_3s$). $O_3s$ is seen to contribute by 40–50% over the Antarctic
       region with greater contribution (45–50%) over the continent with higher elevation than that over
the surrounding ocean (40–45%). The mean stratospheric contribution at the observation site
       Bharati is estimated to be ~47% (~9 nmol mol$^{-1}$), showing that nearly half of $O_3$ at the surface is
       of the stratospheric origin. Mihalikova and Kirkwood, 2013 have estimated a 6–7% occurrence
       rate of tropospheric folds (1-2 folds month$^{-1}$) during summer using radar observations at Troll
       station (72.0° S, 2.5° E, 1275 m amsl). In another recent study also, the enhancement by 20–30
nmol mol$^{-1}$ (67–100% as compared to the climatological mean) is seen in upper tropospheric $O_3$
       above Bharati station due to stratospheric intrusions (Das et al., 2020). Therefore, stratospheric
       intrusions are suggested to transport the $O_3$-rich airmasses to the troposphere, which subsequently
       descend to the surface and get redistributed across the region through horizontal transport. Descent
       of $O_3$-rich airmasses is further discussed in section 3.2.





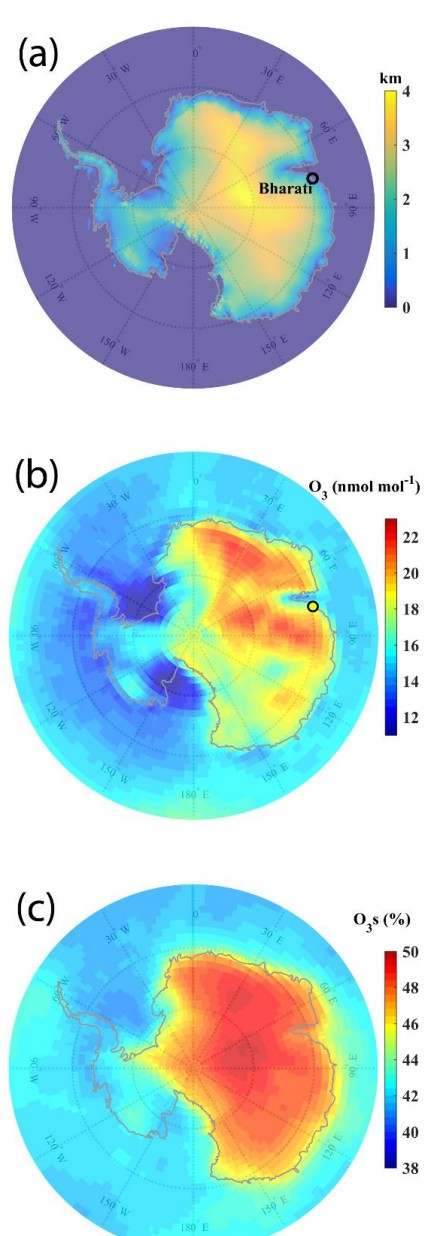

**Figure 1.** (a) Elevation map of Antarctica along with the location of the Indian station Bharati marked by a black circle. (b) Spatial distribution of surface $O_3$ simulated by the EMAC model, averaged over the study period. Colour in the black circle in (b) represents the mean value from





the in situ measurements at Bharati. (c) Percent contribution of stratospheric $O_3$ to the surface $O_3$
derived from the EMAC model during the study period.

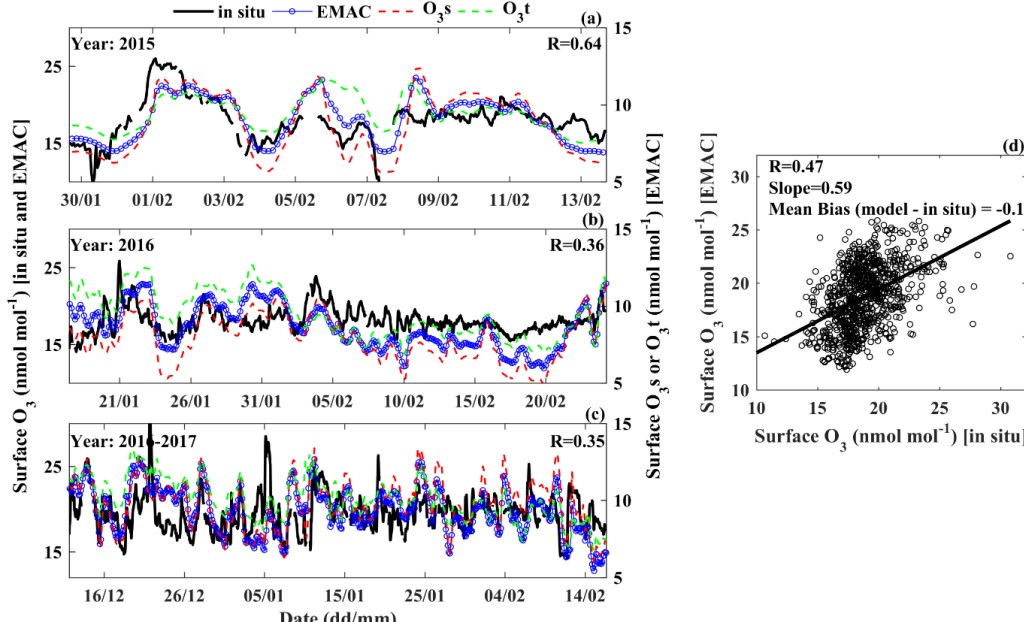

**Figure 2.** Variability in surface $O_3$ (a-c) at Bharati during austral summer of 2015–2017 based on in situ measurements (black) and EMAC simulations (blue). Green and red curves show the absolute stratospheric ($O_3s$) and tropospheric ($O_3t$) contributions to the surface $O_3$. A scatter plot
between in situ measurements and model simulated $O_3$ is shown in (d).

Figure 2a–c shows the variations in surface $O_3$ at Bharati station from in situ measurements and model simulations during the summer seasons of 2015–2017. The mean $O_3$ levels estimated from the model simulations (18.7 ± 3.0 nmol mol$^{-1}$) are in very good agreement with the measurements (18.8 ± 2.3) with negligible bias at this station. Further, the surface $O_3$ level at
Bharati is observed to be similar to an earlier observation (~13–20 nmol mol$^{-1}$) at this station (Ali et al., 2017) and also to other stations in the coastal region of East Antarctica (Fig. S1). The model tends to successfully capture several features of the observed variability (Fig. 2a–c), nevertheless the overall correlation coefficient is 0.47 (Fig. 2d). The comparison for two other coastal stations, Syowa and Arrival Heights during the same study period also shows that the model can reproduce
the summertime $O_3$ levels with small bias and the temporal variability moderately well (Fig. S2–3). The blue and green curves in Fig. 2 show the individual contributions from stratospheric ($O_3s$)




and tropospheric sources (O$_3$t=O$_3$ minus O$_3$s), respectively. Both stratospheric and tropospheric sources are estimated to be contributing nearly equally, 47% and 53%, respectively. Further, the stratospheric and tropospheric O$_3$ at the surface are seen to be strongly correlated (R=0.9; figure not shown) over most of the region mainly due to the mixing of stratospheric and tropospheric airmasses during the transport from the tropopause to the surface. Strong local O$_3$ production (e.g., through NO$_x$ from snow) or direct transport of stratospheric air would decrease the correlation or perturb the variations in O$_3$s and O$_3$t. Overall, similar variability of comparable magnitude in O$_3$s and O$_3$t indicate the absence of strong "local" production or "direct" stratospheric transport to the surface. However, about 50% stratospheric contribution to surface O$_3$ points to significant stratospheric intrusions over the Antarctic region.

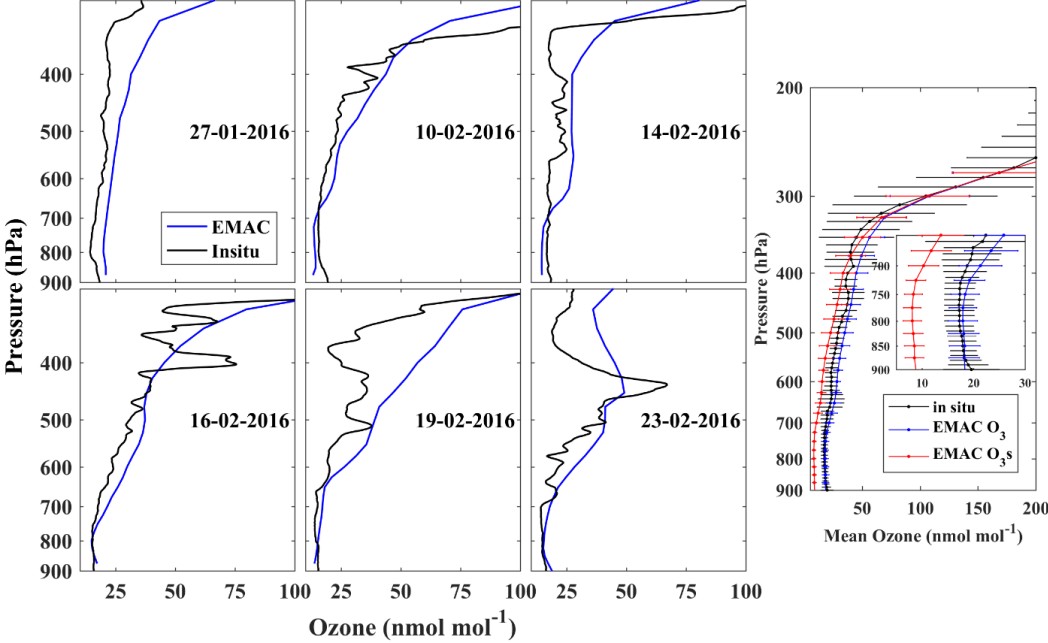

**Figure 3.** Vertical profiles of O$_3$ over Bharati station during a few representative days, based on the in situ measurements (black) and EMAC simulation (blue). The insert on the right shows the mean vertical distribution of O$_3$ and O$_3$s (red) corresponding to 12 profiles during the study period.

Figure 3 shows the comparison of balloon-borne observations of O$_3$ vertical profiles with model simulation over Bharati station in 2016. Out of twelve, six individual representative profiles are shown in the figure. O$_3$ mixing ratios gradually increase with altitude up to the tropopause



(~8.5 km; ~300 hPa), showing also O$_3$ peaks in the middle/upper troposphere during some days. The model successfully captures the mean vertical distribution especially in the lower troposphere (pressure > ~700 hPa) with a mean bias of less than 3 nmol mol$^{-1}$. There is an agreement between the model and observations in the upper troposphere, however, the model overestimates O$_3$ levels (by ~11 nmol mol-1) at the tropopause. Ozonesonde measurements from another station in the region, Davis (68.6° S, 78.0° E), were also compared with the model results for the study period (Fig. S4 and S5). The O$_3$ variability from model results (standard deviation: 3–13 nmol mol$^{-1}$) is comparable or slightly lower than the observed variability in the vertical distribution (950–350 hPa). The O$_3$s contribution is ~45–50% in the lower troposphere (pressure > ~700 hPa) but increases with altitude to 65% at 500 hPa up to 100% at and above the tropopause (~300 hPa). The EMAC model captures both the mean vertical structure as well as some secondary O$_3$ peaks (e.g., 23 February 2016; Fig. 3) in the upper troposphere (~6 km; 450 hPa). However, there are some noticeable differences between model and observations on individual days (e.g., 19 February 2016; Fig. 3). The model limitation in reproducing some features of secondary peaks have been suggested to be due to coarser vertical resolution and the temporal differences (Ojha et al., 2017), and confirmed recently in a study focusing on tropopause folding frequency (Bartusek et al., 2023).

Overall, the model reproduces the observed tropospheric O$_3$ distribution and most of the day-to-day variability in the surface and tropospheric O$_3$. It is to be noted that the performance of global chemistry-climate models is also limited by the parameterization schemes developed for such pristine environments with extreme climatic conditions (e.g., frequent blizzards). Nevertheless, our study fills a gap with respect to the evaluation of the widely applied EMAC model for the Antarctic region and the results may have implications to further improve the model in future studies.

## 3.2 Influences of downward transport on surface O$_3$

Several events of surface O$_3$ enhancements were observed during the study period, as illustrated in Fig. 2. Two such events on 23 February 2016 and 01 February 2015 are investigated in detail to understand the mechanism driving such variability.





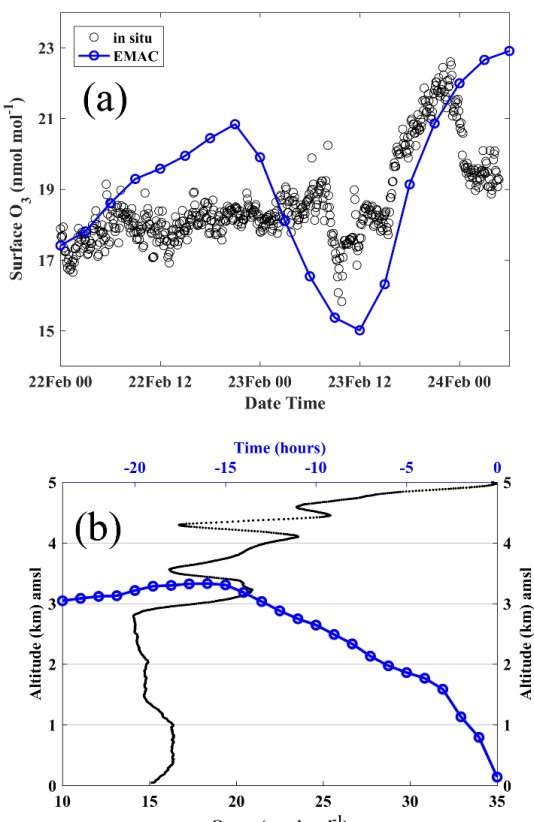

**Figure 4.** (a) surface O$_3$ variations at Bharati station depicting an event of significant O$_3$ enhancement around 23:00 local time on 23 February 2016. (b) Variations in the altitude of airmass (blue) along the backward trajectory with respect to time from the O$_3$ enhancement event. Vertical profile of O$_3$ measured around 11:00 on 23 February 2016 (black).

Figure 4a shows that surface O$_3$ over Bharati was enhanced sharply by ~4 nmol mol$^{-1}$ around 23:00 on 23 February 2016. Backward air mass trajectories show that this air mass originated from ~3 km altitude about 12 hours before the event. The balloon-borne O$_3$ vertical profile obtained at that time (11:00 on 23 February 2016; Fig. 4b) shows the presence of a layer with enhanced O$_3$ (~22 nmol mol$^{-1}$) relative to lower altitudes (~15 nmol mol$^{-1}$). Based on these collocated observations and trajectory simulations, it is suggested that the O$_3$ rich air from this layer descended to the surface over Bharati in ~12 hours with a descent rate of >250 m h$^{-1}$ (0.07 m s$^{-1}$). The estimated descent velocity seems to be consistent with the in situ measured mean vertical wind




speed ($0.09\pm0.29$ m s$^{-1}$) measured at this station during 18–29 January 2016. The O$_3$ enhancement

observed in the upper troposphere (~6 km; see Fig. 3) on 23 February 2016 is associated with a

stratospheric intrusion (Das et al., 2020). The presence of the jet-stream in the vicinity of the

tropopause (~9 km altitude; ~300 hPa) can enhance the turbulence due to strong wind shear

(squared wind shear=$5 \times 10^{-4}$ s$^{-2}$). Along with this turbulence, tropopause oscillations led to the

stratospheric intrusion during 22–23 February 2016 (Das et al., 2020). The presence of similar

surface O$_3$ enhancement events on several other days also (Fig. 2) suggests that this is a periodic

phenomenon that significantly contributes to tropospheric O$_3$ in the region.

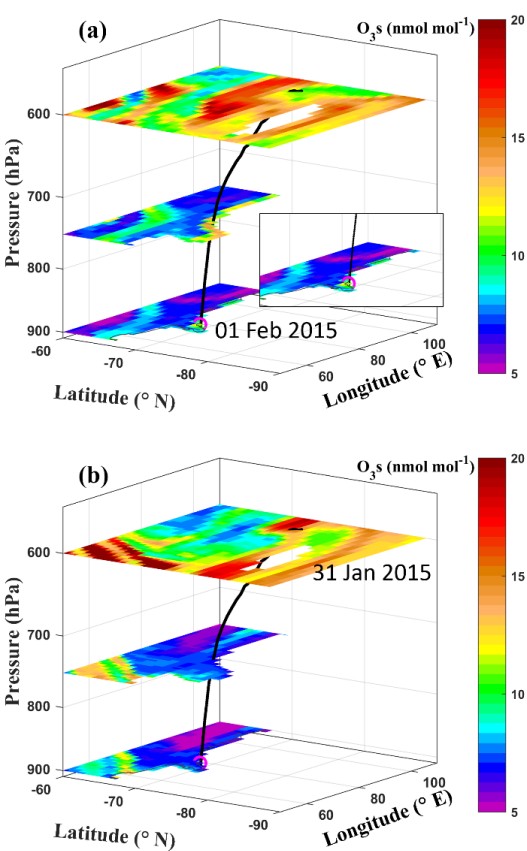

**Figure 5.** Spatial distribution of the stratospheric O$_3$ tracer in the model at different pressure levels

for (a) 01 February 2015 depicting an enhancement at surface, and (b) 31 January 2015 depicting

an enhancement in the upper troposphere. The black curve represents a 24-hour backward air mass



trajectory ending at Bharati station (magenta circle) on 01 February 2015 and originating around
600 hPa on 31 January 2015.

Surface $O_3$ shows a continuous enhancement from about 12–14 nmol mol$^{-1}$ on 31 January
2015 to about 25 nmol mol$^{-1}$ on 01 February 2015 (Fig. 2a). To analyze the influence of transport
from the stratosphere, the spatial distribution of the stratospheric $O_3$ tracer at different pressure
levels is combined with airmass trajectories (Fig. 5). The insert in Fig. 5a shows a zoomed view
of $O_3$s around Bharati station on 01 February 2015. The airmass backward trajectory ending at
Bharati station at the time of the observed enhancement is shown by the black curve. The airmass
is traced back to ~600 hPa (~4 km) one day prior to the observed enhancement (i.e., 31 January
2015) at a lower latitude, where a patch of stratospheric $O_3$ (20 nmol mol$^{-1}$) is simulated by the
model. A clear descent of airmass with a descent rate of ~0.05 m s$^{-1}$ is seen, leading to the
enhancement in surface $O_3$ on 01 February 2015.

The above analysis of two representative events shows that the intrusion of stratospheric $O_3$
followed by descent of $O_3$-rich air can cause a 4–10 nmol mol$^{-1}$ enhancement in surface $O_3$ during
the study period. The result is in line with a continuous increase in $O_3$ and $O_3$s with altitude, as
shown in Fig. 1c and 3. Similar variations of $O_3$t compared to $O_3$s (Fig. 2) indicate significant air
mass mixing during the transport process. $O_3$ enhancement events with similar magnitude were
also observed at the nearby station Zhongshan (69.37° S 76.36° E; Ding et al., 2020; Tian et al.,
2022) and with larger magnitude at South Pole (8–20 nmol mol$^{-1}$; Oltmans et al., 2008) attributed
to transport or $NO_x$ driven cumulative photochemical production, assuming a marginal role of
transport from the stratosphere or free-troposphere (Cristofanelli et al., 2018; Ding et al., 2020).
The occurrence of such $O_3$ enhancement is less evident over the coastal regions compared to the
Antarctic plateau (Jones, 2003). However, substantial contributions of stratosphere-troposphere
exchange were associated with airmass fluxes up to 60 kg m$^{-2}$ d$^{-1}$ (Sanak et al., 1985) using in situ
measurement of Beryllium isotope at Dumont d'Urville station. Based on long-term balloon-borne
measurements and GOES-Chem (Goddard Earth Observing System coupled with Chemistry)
model simulations, Greenslade et al. (2017) also reported large stratosphere to troposphere $O_3$
fluxes (0.50–0.75 × 10$^{17}$ molecules cm$^{-2}$ month$^{-1}$) during summer, which exceed those during
winter (0.25–0.50 × 10$^{17}$ molecules cm$^{-2}$ month$^{-1}$).



**345** **3.3** **Influences of photochemistry on surface O₃**

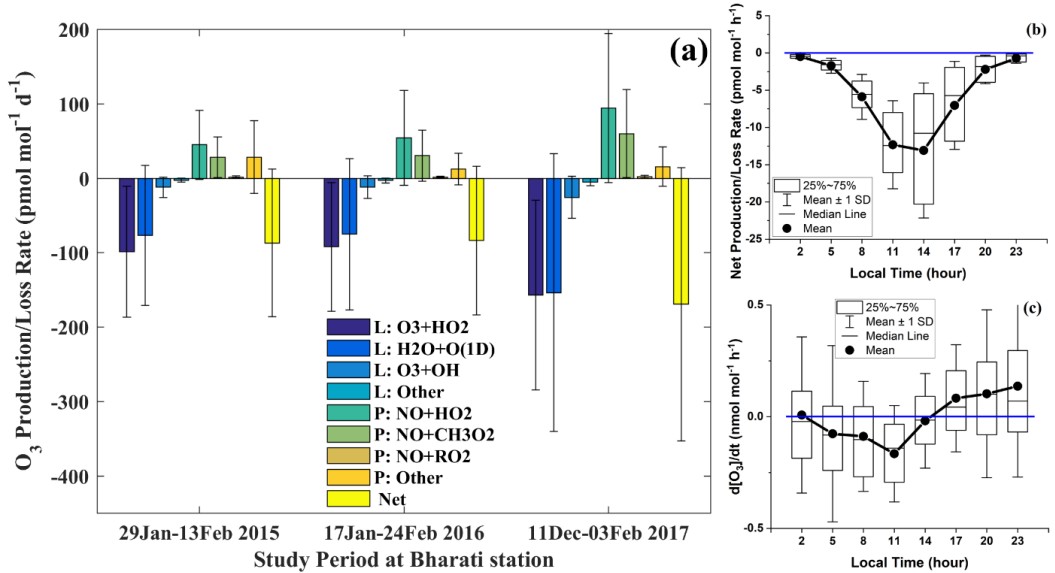

**Figure 6.** (a) Mean production and loss rates of surface $O_3$ through different chemical pathways at Bharati during the study period, (b) diurnal variation of net $O_3$ change due to photochemistry, derived from the EMAC model simulations, and (c) rate of change of surface $O_3$ ($dO_3/dt$) based

**350** on the in situ measurements at Bharati station.

The production and loss rates of $O_3$ through different chemical pathways have been estimated from the EMAC model simulation, and the mean values during the study period are shown in Fig. 6a. Among various production and loss reactions, $O_3+HO_2$ and $H_2O+O(^1D)$ are found to be the dominant $O_3$ loss pathways, whereas $NO+HO_2$ and $NO+CH_3O_2$ are the major $O_3$

**355** production reactions. Overall, the aforementioned chemical losses tend to dominate the production leading to a net photochemical loss in surface $O_3$ at Bharati. Effectively the study region acts as a net chemical sink of $O_3$. Note that loss through $O_3+OH$ and other reactions, and production through $NO+RO_2$ and other reactions are relatively small in magnitude (Fig. 6). Dry deposition over ice and the surrounding ocean is a minor $O_3$ removal mechanism as well. The substantial variability

**360** (large error bars in Fig. 6a) in production and loss terms arises from the diurnal and day-to-day variations. Figure 6b-c shows the mean diurnal variation of net photochemical production or loss rates from the EMAC model and the rate of change of $O_3$ (i.e., $dO_3/dt$) from in situ measurements.



The net loss is relatively high during noontime (11–14 h) and negligibly small after 23:00 and prior to 5:00. In situ measured rate of change, $dO_3/dt$, is negative around 11:00 indicating overall loss which includes the influences of both photochemistry as well as dynamics. The positive rate of change after 17:00 and prior to 5:00 represents an increase in $O_3$ mainly through horizontal or vertical transport as photochemistry is weak under conditions of low solar irradiance.

Despite of being a net photochemical sink of surface $O_3$, it is observed that the levels of $O_3$ are relatively steady or continuous over time (Fig. 2c). We estimated surface $O_3$ fluxes by multiplying the model simulated vertical wind with the $O_3$ concentration at the model level just above the surface. Figure 7b shows the mean $O_3$ flux averaged over the study period. The negative flux represents the number of $O_3$ molecules moving downward (contributing to surface $O_3$) per unit area and per unit of time. A stronger downward flux along the east coast (Fig. 7b) counterbalances the net photochemical $O_3$ loss (Fig. 7a). Assuming a boundary layer height of 500 m, the loss rates integrated over boundary layer are estimated at $2.7 \times 10^{13}$ molecules $m^{-2} s^{-1}$, which is of comparable magnitude to the modelled downward flux (Fig. 7b). $O_3$ and $O_3$ fluxes (mean of fluxes at surface and a level above) correlate negatively (R=-0.3) at Bharati in the EMAC simulation, as shown in Fig. S6a. This is substantiated with a negative correlation of surface $O_3$ with the vertical wind (Fig. S6b), suggesting enhanced $O_3$ during conditions of descent. The results suggest that despite the net chemical sink of $O_3$, the surface $O_3$ is maintained by a flux from above during the summer over the coastal region. The $O_3$ loss through chemistry is counterbalanced by the contribution from dynamics (or vice versa) over East Antarctica during austral summer.





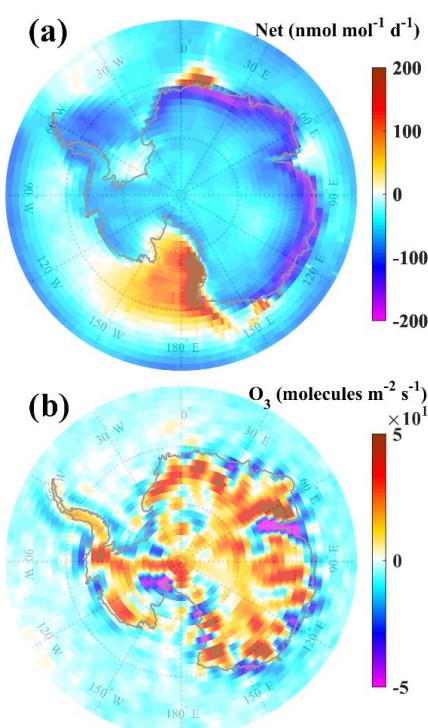

**Figure 7.** Spatial distribution of (a) net rate of change (production minus loss) of surface $O_3$ due
to photochemistry and (b) $O_3$ flux at surface averaged over the study period.

In order to understand whether the $O_3$ photochemical loss over the Bharati station also
prevails over larger regions in Antarctica, we analyze the spatial distribution of net production or
loss rates averaged during the austral summer (Fig. 7a). It is important to note that our simulations
show that the entire Antarctic continent act as sink of $O_3$, in contrast to the previously reported net
$O_3$ production through NO emission from snow (Legrand et al., 2016 and references therein). In
the east coastal Antarctic region $O_3$ loss rates are significantly higher ($\sim$190 pmol mol$^{-1}$ d$^{-1}$)
suggesting that it acts as a relatively strong chemical sink of surface $O_3$. The loss rate is at peak
($\sim$190 pmol mol$^{-1}$ d$^{-1}$) over the east coast, higher by $\sim$100 pmol mol$^{-1}$ d$^{-1}$ compared to adjacent land
and ocean, and it gets further lower ($\sim$50 pmol mol$^{-1}$ d$^{-1}$) much away from the coast. Note that
model simulated mean OH and NO are in the range of 0.05–0.5 × 10$^6$ molecules cm$^{-3}$ and 0.5–10
pmol mol$^{-1}$, respectively, over entire the Antarctic region which is in line with earlier
measurements at the west coast (OH mean: 0.11 × 10$^6$ molecules cm$^{-3}$, ranging <0.1–0.9 × 10$^6$



molecules $cm^{-3}$; NO: estimated value of 5 pmol $mol^{-1}$) by Jefferson et al., 1998 and Bloss et al.,
2010, but lower (almost 5 times) than those measured during OPALE campaign (OH mean: 2.1 ×
$10^6$ molecules $cm^{-3}$, ranging <0.8–6.2 × $10^6$ molecules $cm^{-3}$; NO: 5–70 pmol $mol^{-1}$; Kukui et al.,
2012).

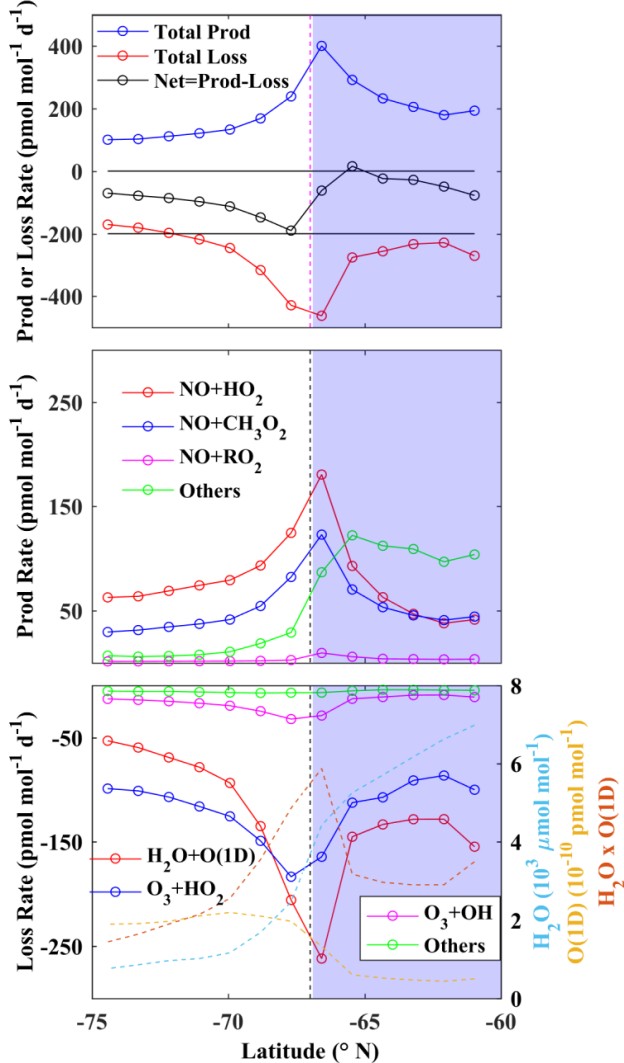

**Figure 8.** Latitudinal variation of production, loss and net rates of changes of surface $O_3$ averaged
along the east coastal longitudinal band of 15–130° E during January 2017. The blue areas
represent the ocean environment and vertical dashed line shows the approximate coast line.





The net $O_3$ loss rate (Fig. 7a) is found to be lower over land than over ocean and is highest along the east coast. We further considered 6 grids at both sides of the coast line and averaged over the longitude range 15–130° E (i.e., East Antarctica). The variations in average production, loss and net rates with latitude are shown in Fig. 8. Since the latitude corresponding to different grids at 15–130° E are different, latitudes shown on the x-axis represent average latitudes. Thus, as we proceed from left to right (lower latitude to higher latitude) in Fig. 8, we move from land to ocean.

From Fig. 8a, it is clearly seen that the $O_3$ production as well as loss are maximum near the coast. Since loss dominates over the production, the net rate is negative with ~190 pmol mol$^{-1}$ d$^{-1}$. Figure 8b and c represent changes in different production and loss pathways across the coast. The photolytic $O_3$ loss, followed by $H_2O+O(^1D)$, is found to be the dominating loss process peaking at ~300 pmol mol$^{-1}$ d$^{-1}$ along the coast. The reason for the peak loss rate at coast is related to the opposite latitudinal gradients in $H_2O$ and $O(^1D)$ (see Fig. 8c; right axis).  $H_2O$ is substantially higher (~6000 μmol mol$^{-1}$) over ocean but much lower (1000 μmol mol$^{-1}$) in the drier atmosphere above continent. In contrast, $O(^1D)$ is higher ($2 \times 10^{-10}$ pmol mol$^{-1}$) over the continent primarily due to intense solar insolation at higher elevation and over the bright ice surface. Therefore, latitudinally opposite variations of $H_2O$ and $O(^1D)$ lead to a relative maximum in $H_2O+O(^1D)$ near the coast. We also note that there is significant $O_3$ production over the ocean due to reactions other than the three primary reactions of peroxy radicals ($HO_2$, $RO_2$, $CH_3O_2$) with NO.

Under the prevailing relatively strong $O_3$ sink along the east coast, the mean $O_3$ level during summer is maintained by the downward flux of $O_3$ from the stratosphere.

### 3.4    Diurnal variation of surface $O_3$ at Bharati

Considering day-to-day variability, including enhancement events governed by stratospheric influence, $\Delta O_3$ is computed by subtracting the running mean $O_3$ (288 points; 5 min interval—daily running mean) from the observed $O_3$. Figure 9a shows the mean diurnal variation of $\Delta O_3$ during the 18 January–23 February 2016 period for which measurements of horizontal wind at surface were also available. Surface $O_3$ exhibits a diurnal variation, being relatively low during the afternoon (15:00 local time) and relatively high during nighttime (Fig. 9) with a diurnal amplitude of ~1.2 nmol mol$^{-1}$. Figure 9b shows the wind rose color coded with $\Delta O_3$ mixing ratios. Sunlight at Bharati is abundant during summer and the land-sea thermal contrast explains the typical diurnal



change in the wind direction under normal meteorological conditions, i.e., excluding blizzards and
       snow storms. Figure S7 shows time series of the wind direction and surface $\Delta O_3$, depicting the
       link between $O_3$ and the wind direction. Due to higher $O_3$ over the eastern Antarctic land regions,
       winds from that sector transport the $O_3$-rich air to the Bharati station causing enhanced $O_3$ mixing
       ratios. $O_3$ is higher when wind is parallel to the coast (easterly; wind direction ~90°) or from the

land (wind direction: 90–240°). Under calm wind condition, the influence of transport is minimal
       and photochemical loss is more pronounced. When the wind is weak and from the ocean (wind
       direction: 30–90° N), $O_3$ levels are lower due to dilution by mixing with air from the oceanic
       sector. The $O_3$ diurnal variation is also closely linked with the vertical wind. Based on limited in
       situ measurements of the vertical wind at the surface during 18–29 January 2016, the mean diurnal

variation of vertical wind (w) along with $\Delta O_3$ is shown in Fig. 9c. Downdrafts and stronger
       updrafts (up to ~0.4 ms$^{-1}$) are seen during nighttime (or lower solar zenith angle; 20:00–07:00) and
       daytime (08:00–19:00), respectively. Higher $O_3$ during nighttime is associated with downdrafts
       and $O_3$ mixing ratios are reduced with increasing updraft intensity. The EMAC model shows
       limitations in reproducing the observed diurnal variation likely because of coarse resolution

averaging out the topography and mesoscale dynamics.



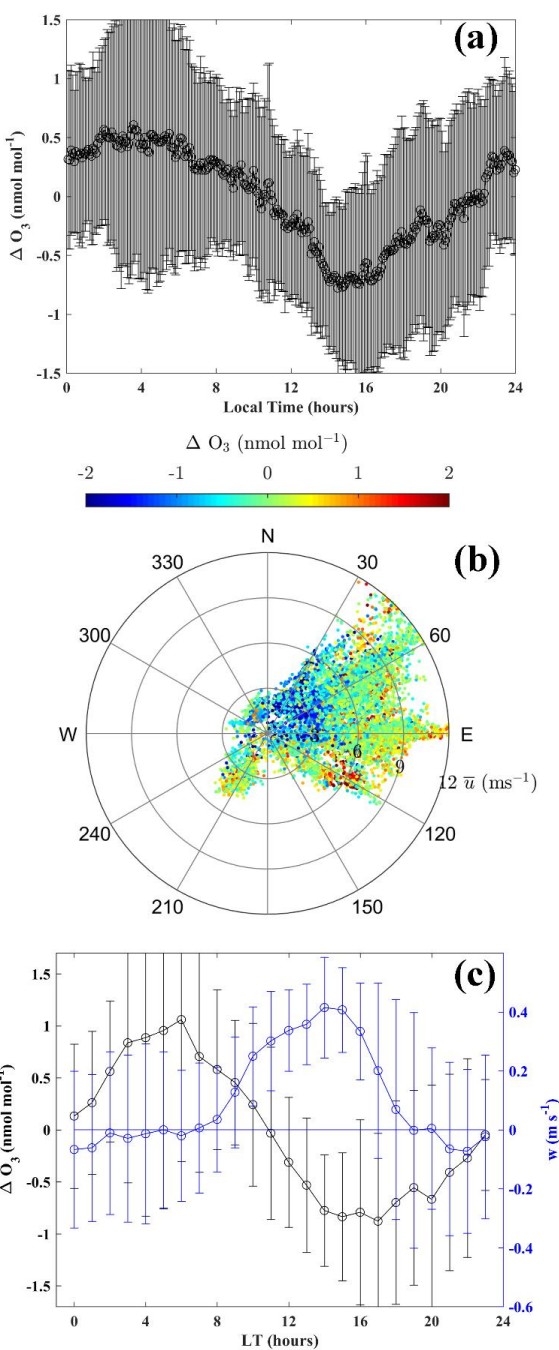

**Figure 9.** (a) Diurnal variation of $\Delta O_3$, (b) wind rose color coded with $\Delta O_3$, and (c) variation in collocated vertical wind and $\Delta O_3$ at Bharati during the austral summer of 2016.



Diurnal patterns with an amplitude ranging from ~0.2–2 nmol mol$^{-1}$ were reported at coastal (Syowa and McMurdo) and inland Concordia (75° S; 123° E; 3220 m above sea level) stations (Ghude et al., 2006; Legrand et al., 2009). However, such a pattern is absent over the South Pole (Oltmans, 1981). Interestingly, photochemical production during the morning hours (05:00–11:00) due to the $NO_x$ released from snow was followed by a reduction due to an increase in boundary

layer height (200 ±100 m) at the inland station Concordia (Legrand et al., 2009; 2016). Shallow convective boundary layers (less than 300 m) were reported over the Antarctic Plateau region by Mastrantonio et al. (1999). Unlike these studies, we did not observe photochemical $O_3$ production nor a clear signature of changes in $O_3$ transport across the top of boundary layer from our ozonesonde measured $O_3$ profiles over Bharati station. Therefore, the diurnal patterns of $O_3$ over

coastal Antarctica are found to be different than those over the inland region, mainly due to differences in meteorological conditions and the concentrations of precursor gases.

### 3.5   Absence of signature of halogen chemistry

      Reactive halogens (e.g., iodine, bromine) have been shown to deplete $O_3$ in the boundary layer over the Antarctic region (Barrie et al., 1988; Oltmans and Komhyr, 1976). However,

ground-based remote sensing observations found very low concentrations of iodine oxide (~0.3±0.1 pmol mol$^{-1}$) in the boundary layer over Bharati station during the study period (Mahajan et al., 2021) and no clear sign of $O_3$ depletion was observed.

      Satellite (SCIAMACHY—SCanning Imaging Absorption spectroMeter for Atmospheric CartograpHY and OMI—Ozone Monitoring Instrument) observations also show lower monthly

mean iodine monoxide (IO) columnar density (0–1×10$^{12}$ molecules cm$^{-2}$) over Bharati compared to west Antarctica (figure not shown). This is consistent with previous studies (e.g., Schönhardt et al., 2012) showing relatively low IO over east Antarctica and the adjacent ocean ($\leq 0.7 \times 10^{12}$ molecules cm$^{-2}$) compared to west Antarctica (~$1.5 \times 10^{12}$ molecules cm$^{-2}$) during summer season (December-January-February 2004–2009).

Bromine (Br) driven $O_3$ depletion events, resulting into BrO, are less frequent over the Antarctic region compared to the Arctic region due to differences in springtime surface temperatures (Tarasick and Bottenheim, 2002). However, large $O_3$ depletion events were observed



at Neumayer (70.62° S, 8.37° W; 42 m amsl) during the late winter (July to September), likely due to stronger BrO episodes from the larger sea ice coverage around the site (Legrand et al., 2009).

Analysis of BrO from OMI possibly indicates an $O_3$ depletion event on 7 February 2015 at Bharati where BrO was enhanced, ~$9.2 \times 10^{13}$ molecules cm$^{-2}$ with lower $O_3$ (~7 nmol mol$^{-1}$), marked by red rectangle in Fig. S8. Except for this event, BrO remained below $8 \times 10^{13}$ molecules cm$^{-2}$ around Bharati station (±0.5° latitude/longitude) during the study. $O_3$ depletion was also not seen at Syowa (Fig. S9) during the study period. The coastal region of east Antarctica exhibits slightly higher

values of BrO (~$7 \times 10^{13}$ molecules cm$^{-2}$) compared to the ocean and land regions (4–6$\times 10^{13}$ molecules cm$^{-2}$). However, it is low (4–8 $\times$ $10^{13}$ molecules cm$^{-2}$) during December–February (2004–2009) compared to the levels during September–November (5–10 $\times$ $10^{13}$ molecules cm$^{-2}$) over the Antarctic region (Schönhardt et al., 2012). The impact of Br chemistry on surface $O_3$ is suggested to be weaker along the east coast of Antarctica (Dumont d'Urville and Syowa) in

contrast to western coastal Antarctica as observed over the Neumayer and Halley (75.55° S, 26.53° W, 30 m amsl) stations (Legrand et al., 2016). Nevertheless, simultaneous measurements of $O_3$ and halogen species including BrO are desirable to quantify the role of halogen chemistry over eastern Antarctica.

### 3.6    Surface $O_3$ during winter

To take into account the seasonality of $O_3$ at the surface, the wintertime distribution is shown in Fig. S10. Mean surface $O_3$ level is higher during winter (20–32 nmol mol$^{-1}$) compared to the summer (11–23 nmol mol$^{-1}$), in line with the reported seasonality in the literature (Legrand et al., 2009). Figure S10 reveals three low-$O_3$ patches over the coastal oceanic region. One is close to Bharati station, however, we do not have observations during wintertime for comparison. Model

simulations suggest that surface $O_3$ is composed of 63–67% $O_3$s of stratospheric origin during winter (Fig. S10b), which is significantly higher than during austral summer. The probability of downward transport from the stratosphere during winter, also associated with a lower altitude of the tropopause, is larger (Kumar et al., 2021). Comparison of surface $O_3$ at Syowa (69.00 °S; 39.58 °E; not shown here) shows that the model captures the variability with R=0.3 and a negative bias

of ~5 nmol mol$^{-1}$. The model performance seems to be better during summer, indicative of limitations to reproduce the mean $O_3$ concentrations and the variability during winter. Analysis of the $O_3$ budget suggests a small net loss of $O_3$ by 10–25 pmol mol$^{-1}$ d$^{-1}$ over the oceanic region and



close to zero (<5 pmol mol$^{-1}$ d$^{-1}$) over the Antarctic continent (Fig. S11). To study this in greater detail, we highly recommend to conduct continuous wintertime measurements of O$_3$, its precursors

including halogens over Bharati during winter season.

## 4    Summary

Ground- and balloon-borne O$_3$ measurements have been conducted over the Indian station Bharati at the east coast of Antarctica during the austral summers of 2015–2017. The observations have been used to evaluate the performance of the global chemistry-climate model EMAC over

this part of the world. A comprehensive analysis of observations and model simulations provided significant insights into the dynamical and photochemical processes affecting surface O$_3$ and its variability. The main results are:

1. Surface O$_3$ levels over the Indian station Bharati at the eastern coastal Antarctica have been observed to be ~19 nmol mol$^{-1}$ with a small variability of ~2 nmol mol$^{-1}$ during austral summer.

While similar levels prevail over the east coast, O$_3$ is typically higher over land at higher elevation. EMAC model successfully reproduced the observed mean levels with negligible bias over this unique environment and also captured the temporal variability (R=0.5). In particular, the model successfully reproduced some events during which O$_3$ was enhanced. Analysis of the stratospheric O$_3$ tracer in the model suggests that 40–50% of surface O$_3$ is of stratospheric origin

with larger fractions over the higher elevation regions in Antarctica.

2. The model successfully reproduced the mean vertical distribution of O$_3$ over Bharati observed by balloon-borne soundings. Detailed analysis combining the balloon profiles, model tracers, and air mass trajectories shows that downward transport caused the observed events during which O$_3$ was enhanced.

3. Along the east coast of Antarctica, including Bharati station, photochemistry acts as a relatively strong sink of surface O$_3$ (~190 pmol mol$^{-1}$ d$^{-1}$) when compared to adjacent land and ocean regions. Chemical loss through O$_3$ photolysis (followed by H$_2$O+O($^1$D)) and O$_3$+HO$_2$ dominates over the major production (through NO+HO$_2$ and NO+CH$_3$O$_2$). Reverse latitudinal gradients between H$_2$O and O($^1$D) lead to maximum O$_3$ loss at the coastal region. The

continuous chemical loss is found to be counterbalanced by downward O$_3$ transport from above. The findings show the intertwined roles of dynamics and photochemistry that govern the O$_3$





variability over east Antarctica, and maintaining significant $O_3$ levels despite the absence of local precursor sources.

4. In addition to the role of photochemistry, the diurnal variation of $O_3$ at Bharati was found to
correlate with the diurnal wind changes. Surface $O_3$ varied with a diurnal amplitude of 1.2 nmol $mol^{-1}$, with the higher levels occurring when the wind blew parallel to the coast or from land regions. In addition, up- and downdrafts also play a role in the diurnal variation.

Our observations during austral summer over three years complement available data e.g.
from eastern coastal Antarctica. The observations, besides revealing diurnal and day-to-day variability, helped in evaluating the performance of a global chemistry-climate model over this unique, pristine environment. The study provides valuable insights into the complementary roles of photochemistry and dynamics in governing $O_3$ and its variability over Antarctica. In view of increasing anthropogenic activities and the changing climate, monitoring of $O_3$ and related species
(NO, $NO_2$, CO, VOCs and halogens) is needed.

*Code availability.* The Modular Earth Submodel System (MESSy) is continuously further developed and applied by a consortium of institutions. The usage of MESSy and access to the
source code is licensed to all affiliates of institutions that are members of the MESSy Consortium. Institutions can become a member of the MESSy Consortium by signing the MESSy Memorandum of Understanding. More information can be found on the MESSy Consortium Website (http://www.messy-interface.org, last access: 04 July 2023). The code presented here has been based on MESSy version 2.55 and is available as git commit #a5bd54d5b in the MESSy repository.


*Data availability.* Measured ozone and EMAC simulated fields shown in the figures can be obtained from the website of Space Physics Laboratory (https://spl.gov.in/SPL/index.php/spl-metadata/104-spl/550-trace-gases-metadata) or from the direct link, https://spl.gov.in/SPL/images/SPL-METADATA/Ozone_Bharati_Antarctica_Summer_2015-
2017.xlsx.



*Author contributions.* I.A. Girach conceptualized and designed the study, performed measurements and analysed the datasets. K.V. Subrahmanyam, Koushik N., Mohammed Nazeer M., N.V.P. Kiran Kumar contributed in the measurements. A. Pozzer preformed the model simulations. N. Ojha, A. Pozzer, P.R. Nair, S.S. Babu and J. Lelieveld helped I. A. Girach in the analysis and interpretation of the results. I. A. Girach wrote the manuscript and all the co-authors contributed to the review and editing.

*Declaration of competing interest.* At least one of the (co-)authors is a member of the editorial board of Atmospheric Chemistry and Physics.

*Acknowledgements.* We gratefully acknowledge the organiser, Centre for Polar and Ocean Research (NCPOR), Goa, Ministry of Earth Sciences, India for providing the opportunity to participate in the 34th, 35th and 36th Indian Scientific Expedition to Antarctica (ISEA). We also acknowledge the leaders of Bharati Station and Voyage for providing necessary support for the smooth conduct of experiments at Bharati station. We are really thankful to Mr. Santosh Muralidharan, Space Physics Laboratory; Mr. Brijesh Desai, Laboratory in-charge of Bharati station during 35th expedition and other expedition members of 34th, 35th and 36th ISEA for their help during the field measurements. We are also thankful to India Meteorological Department (IMD) for providing meteorological observations, hydrogen gas cylinders for balloon ascents and for the help during balloon launches. The EMAC model simulations have been performed at the German Climate Computing Centre (DKRZ). Surface ozone observations at Antarctic stations (South Pole, United States; Arrival Heights, New Zealand; Marambio, Argentina; Syowa, Japan) were obtained from the newly established World Data Centre for Reactive Gases (WDCRG), WMO's GAW (Global Atmosphere Watch; World Meteorological Organization) programme (https://ebas.nilu.no/ and https://ebas-data.nilu.no/Default.aspx). Vertical $O_3$ profiles measured at Davis station were obtained from https://woudc.org/data/explore.php. We highly acknowledge teams of researchers who made ozone measurements at various Antarctic stations and made them available publically. We also acknowledge the NOAA Air Resources Laboratory (ARL) for providing air mass trajectory from the HYSPLIT transport and dispersion model from their READY website (http://www.arl.noaa.gov/ready.php).



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
