# Peer review of "Influences of downward transport and photochemistry on surface ozone over East Antarctica during austral summer: in situ observations and model simulations"

_EGUsphere, 2023_

## Author Comment (AC1)

**Response to comments of Reviewer-1**

Overall Comment: The paper presents new observations of ozone over the East Antarctic Bharati station and interprets them in terms of stratospheric-tropospheric transport using a chemistry-climate model and trajectories back from soundings. It will be of interest to readers who expect to see the data online.

Response: We thank the reviewer for constructive comments, which have helped us to improve the manuscript significantly. The manuscript has been revised by addressing all the comments. Point-by-point responses to the comments are given below in blue fonts. The discussion added/updated in the manuscript is presented by red text.

Main recommendation: post the ozonesonde data- have you sent this to woudc.org and registered the site with them? Your institutional link should be a primary location but the ozone profile user community will look for the data at woudc.org. Thank you.

Response: To place the data on woudc.org, GAW ID for the station need to be created after the activation of registration, which needs recommendation by the National Focal Point. We have applied but the registration could not be activated yet. We will send the ozonesonde data to woudc.org as soon as the registration gets active. Nevertheless, all data has been made available on our institutional website.

Minor Comments:

Comment 1: Although this reference is not new, it includes examples of chemical processes at the snow-ice interface that could be relevant to your study: Biogeochemical Cycles and Ice Cores, NATO ASI Series I-30, ed. R. J. Delmas. ISBN 3642511740; ISBN-13 978-3642511745. Did you consider gases interacting with ozone that may come from snow-pack during the summer?

Response 1: We have incorporated snow-air interaction in the simulation. We further improved the simulation with corrected background lighting $NO_X$ and by including bromine emission through snow-air interactions, and therefore we revised all the figures. Since the results did not change significantly, there is no change in conclusions or discussions.

Comment 2: Lines 280-286 – Vague. How would blizzards, etc, affect the observations? You imply model improvements need to be made to give a more accurate simulation. Be more specific.

Response 2: We intend to say that global models have limitation in simulating extreme events like blizzards due to application of generalised global schemes of parameterisation and coarse grid resolution (in this case $1° \times 1°$). Depletion of surface ozone is observed over Antarctica during blizzards because blowing snow is a source of sea salt aerosol and subsequently bromine (Jones et al., 2009; Ali et al., 2017). Therefore, if such events are not well captured, it is likely that model could show mismatch with observations. To clarify, following statement is included in the revised manuscript.

Lines 295–298: Note that depletion of surface ozone was observed over Antarctica during blizzards as blowing snow, which is a source of sea salt aerosols and subsequently bromine, could deplete ozone (Jones et al., 2009; Ali et al., 2017).

Comment 3: Lines 505-510. The R for Syowa disagreement suggests that the model is not doing well in winter at al. What does this mean? Reasons?

Response 3: Yes, model in its present configuration (section 2.2) shows limitation in capturing the observed variability at Syowa. The reasons for such discrepancies are not fully understood, and detailed study with more winter time observations and simulation would be needed to find the reasons for model's wintertime limitation and possible improvement in the model. This is mentioned explicitly in the revised manuscript.

Lines 528-530: Further studies are needed to understand and rectify the factors causing greater bias in the model during winter.

---

## Author Comment (AC2)

**Response to comments of Reviewer-2**

General comment-1: Girach et al. analyze surface ozone measurements at the Indian East Antarctic Bharati station in comparison with simulations from the chemistry climate model EMAC. The model is used to discriminate the role of subsidence from the stratosphere versus photochemical production in the troposphere or at surface level.

My main criticism is that the model description is not clear enough if or how chemical processes at the snow surface are included, in particular NOx emissions from the snow pack halogen/bromine chemistry, and the dry deposition of ozone. Even if it is argued that these processes may not be relevant for this particular study, at least it should be made clearer in Secton 2.2 which chemical processes are considered by the present EMAC simulations. Relevant EMAC sub-models seem to be available for polar bromine chemistry (Falk and Sinnhuber, 2018, https://doi.org/10.5194/gmd-11-1115-2018) and this latter study also investigated the uncertainty in ozone dry deposition velocities (as did other studies).

Response: We thank the referee for the valuable comments. We have incorporated snow-air interaction in the new simulation and revised all the figures. We have now provided better description of the model including chemical processes at the snow surface. Following description is added in the section 2.

Lines 193–200: The model includes emissions of bromine from sea spray following the approach of Kerkweg et al. (2008), and important heterogeneous reactions involving bromine (e.g., liquid phase reactions of HOBr + HBr → $Br_2$ + $H_2O$) are included via the AERCHEM subroutines (Rosanka et al., 2023) in the GmXe submodel (Pringle et al., 2010). With the ONLEM submodel, the air-snow subroutine are activated (Falk and Sinnhuber, 2018), which include the bromine release on sea-ice and snow-covered surface, based on the scheme of Toyota et al. (2011). Beside the bromine release, no $NO_x$ release is included by the deposition of $O_3$. Note that $NO_x$ and HONO emissions from snowpack (Honrath et al., 2002; Bond et al., 2023) are not incorporated in the model.

General comment-2: Overall the manuscript is well written and will be an interesting contribution to the literature, investigating the processes that contribute to ozone variability and trends in a data sparse region. I recommend publication in Atmos. Chem. Phys. after the following comments are taken into account.

Response: We thank the reviewer for constructive comments, which have helped us to improve the manuscript significantly. The manuscript has been revised by addressing all the comments. Point-by-point responses to the comments are given below in blue fonts. The discussion added/updated in the manuscript is presented by red text.

Specific comments:

Comment-1: l.217: The statement on the tropopause fold occurrence frequency is somewhat disconnected and it is not clear how the conclusion can be made that the stratosphere to troposphere ozone flux is dominated by tropopause folds in contrast to slow subsidence through the tropopause.

Response 1: Contribution of stratospheric ozone at surface is combined effect of tropopause folds as well as slow subsidence trough tropopause and not distinguishable here. Therefore, it cannot be

concluded from the present study that stratosphere to troposphere ozone flux is dominated by tropopause folds. We have briefly mentioned this in the revised manuscript.

Lines 230–233: In addition, gradual subsidence through the tropopause can also contribute to stratospheric ozone transport into the troposphere.

Comment-2: l.250: O3s and O3t are correlated "due to mixing during the transport from the tropopause". This statement confuses me. So does that mean O3s and O3t do not really represent the stratospheric and tropospheric contributions any more, but rather a mixture of the two? How useful are they then as a diagnostic??

Response 2: Despite they show good correlation, O3s and O3t quantify the stratospheric/tropospheric contributions. Atmospheric dynamical processes do not influence selectively only O3s or O3t as basically they are identical molecules. Therefore, they can mix in the troposphere and they can exhibit strong correlation mainly due to dynamical processes. When there is fresh production or direct influence of stratospheric intrusion, correlation would break/decline over the duration of production/intrusion. Therefore, they are good diagnostics not only for quantifying the stratospheric/tropospheric contribution but also to identify direct or immediate influence of tropospheric production or stratospheric intrusion.

Comment-3: l.252: "Strong local O3 production (e.g., through NOx from snow)": again, is O3 production through NOx from snow included in the EMAC simulations? If not, is this some kind of circular reasoning? If it is included, would be good to give a few more details.

Response 3: As the referee correctly pointed out, no NOx emissions is included in the model. However, no significant change is seen in the result. We have revised the sentence.

Line 264: Direct transport of stratospheric air or local production of $O_3$ would decrease the correlation.

Comment-4: l.285: "to further improve the model in future studies": how? Can you give some hints what may need to be improved?

Response 4: As the referee correctly pointed out, not all necessary air-snow interactions have been included in the model. One example is the $NO_x$ and HONO emissions, which are results of nitrate photolysis within the snowpack (Honrath et al. 2002, Bond et al. 2023).

Comment-5: Fig. 6b/c: The modelled net chemical tendencies (up to around 15 pmol/mol/h) are 1 order of magnitude smaller than the mean observed O3 tendencies (on average around 0.2 nmol/mol/h during morning and noon). Are the observed O3 tendencies in 6c not statistically significant? Or are there removal processes missing?

Response 5: Tendencies shown in Figure 6b are purely due to chemistry (net chemical production or loss). Whereas in figure 6c, it is from the observations which includes changes in ozone due to transport (horizontal and vertical) as well as deposition losses. Therefore, large difference is expected. The aim of showing these two together is that negative tendencies in the observed ozone during noontime is in line with chemical losses. Amplitude of mean tendency shown in figure 6c is 0.3 which is comparable to variability at any given hour of the day. Therefore, diurnal patterns on different days may slightly differ from the mean picture shown in figure 6c.

We have revised the text for better clarity as followings.

Lines 378–382: In situ measured rate of change, $dO_3/dt$, is negative around 11:00 indicating overall loss which includes the influences of both photochemistry, dynamics and deposition losses. Since amplitude of mean $dO_3/dt$ in figure 6c is 0.3 nmol mol$^{-1}$ h$^{-1}$, which is comparable or smaller than variability at any given hour of the day. Therefore, diurnal patterns on different days might vary from the mean picture.

Comment-6: l.423: can you give us some idea what the "other" ozone production includes?

Response 6: Here, in figure 8b (green curve), "other" includes all possible reactions excluding reaction of NO with peroxy radicals ($HO_2$, $RO_2$, $CH_3O2$) through which ozone is produced. This could be also reactions of $HO_2$ with other peroxy radicals formed after long range transport (e.g. reaction of acetyl peroxy radical with $HO_2$: $CH_3CO_3 + HO_2$). Since this is estimated by subtracting three production terms from the total production, we could not get insight about other individual production terms.

Minor comments:

Comment-7: l.65: "increasing trend (<0.2 nmol/mol/y)": the number refers to a trend, not a trend increase. So either give some numbers how the trend is increasing or delete the word "increasing" in this context. Moreover, the "<" means this is an upper limit for the trend; better give a lower limit if available.

Response 7: We have excluded "increasing" term and provided the range (0.08–0.13 nmol/mol/y over Syowa, Arrival Heights, Neumayer, South Pole) of observed positive trends.

Lines 65–66: A positive trend (0.08–0.13 nmol mol$^{-1}$ y$^{-1}$ over Syowa, Arrival Heights, Neumayer, and South Pole) in surface $O_3$ has also been reported from Antarctica.

Comment-8: l.209: I suppose "Summit" is on Greenland? Why is this included here? If Summit should be included, please include lat/lon and/or geographic reference.

Response 8: Yes, it is on Greenland, and was mistakenly included. Now we have removed this in the revised manuscript (Line 218).

Comment-9: Fig.1 Caption: suggest to include explicitly the time period shown

Response 9: It is average over the study period (29 January–13 February 2015, 17 January–24 February 2016, and 11 December 2016–16 February 2017), now mentioned in the figure caption (Lines 238–239).

Comment-10: Fig.2 Caption: suggest to mention that O3s and O3t are on a different scale than O3. (It took me a while before I realized that, couldn't make sense of it before)

Response 10: Now we have revised the figure caption by mentioning the different scale for O3s and O3t (Lines 246–247).

References:

Bond, A. M. H., Frey, M. M., Kaiser, J., Kleffmann, J., Jones, A. E., and Squires, F. A.: Snowpack nitrate photolysis drives the summertime atmospheric nitrous acid (HONO) budget in coastal Antarctica, Atmos. Chem. Phys., 23, 5533–5550, https://doi.org/10.5194/acp-23-5533-2023, 2023.

Falk, S. and Sinnhuber, B.-M.: Polar boundary layer bromine explosion and ozone depletion events in the chemistry–climate model EMAC v2.52: implementation and evaluation of AirSnow algorithm, Geosci. Model Dev., 11, 1115–1131, https://doi.org/10.5194/gmd-11-1115-2018, 2018.

Honrath R.E, Lu Y, Peterson M.C, Dibb J.E, Arsenault M.A, Cullen N.J, Steffen K, Vertical fluxes of NOx, HONO, and HNO3 above the snowpack at Summit, Greenland, Atmospheric Environment, https://doi.org/10.1016/S1352-2310(02)00132-2, 2002.

Jones, A. E., Anderson, P. S., Begoin, M., Brough, N., Hutterli, M. A., Marshall, G. J., Richter, A., Roscoe, H. K., and Wolff, E. W.: BrO, blizzards, and drivers of polar tropospheric ozone depletion events, Atmos. Chem. Phys., 9, 4639–4652, https://doi.org/10.5194/acp-9-4639-2009, 2009.

Kerkweg, A., Jöckel, P., Pozzer, A., Tost, H., Sander, R., Schulz, M., Stier, P., Vignati, E., Wilson, J., and Lelieveld, J.: Consistent simulation of bromine chemistry from the marine boundary layer to the stratosphere – Part 1: Model description, sea salt aerosols and pH, Atmos. Chem. Phys., 8, 5899–5917, https://doi.org/10.5194/acp-8-5899-2008, 2008.

Pringle, K. J., Tost, H., Message, S., Steil, B., Giannadaki, D., Nenes, A., Fountoukis, C., Stier, P., Vignati, E., and Lelieveld, J.: Description and evaluation of GMXe: a new aerosol submodel for global simulations (v1), Geosci. Model Dev., 3, 391–412, https://doi.org/10.5194/gmd-3-391-2010, 2010.

Rosanka, S., Tost, H., Sander, R., Jöckel, P., Kerkweg, A., and Taraborrelli, D.: How non-equilibrium aerosol chemistry impacts particle acidity: the GMXe AERosol CHEMistry (GMXe–AERCHEM, v1.0) sub-submodel of MESSy, EGUsphere [preprint], https://doi.org/10.5194/egusphere-2023-2587, 2023.

Toyota, K., McConnell, J. C., Lupu, A., Neary, L., McLinden, C. A., Richter, A., Kwok, R., Semeniuk, K., Kaminski, J. W., Gong, S.-L., Jarosz, J., Chipperfield, M. P., and Sioris, C. E.: Analysis of reactive bromine production and ozone depletion in the Arctic boundary layer using 3-D simulations with GEM-AQ: inference from synoptic-scale patterns, Atmos. Chem. Phys., 11, 3949–3979, https://doi.org/10.5194/acp-11-3949-2011, 2011.